# Modern Pollen Analysis in the Estuary Habitats along the Coast of Dhofar (Sultanate of Oman)

**Cristina Bellini** [1,*], **Francesco Ciani** [1] , **Lia Pignotti** [1,2], **Riccardo Maria Baldini** [1,2], **Tiziana Gonnelli** [1] and **Marta Mariotti Lippi** [1,2]

[1]  Dipartimento di Biologia, Università di Firenze, Via G. La Pira, 4, 50121 Florence, Italy
[2]  Centro Studi Erbario Tropicale, Università di Firenze, Via G. La Pira, 4, 50121 Florence, Italy
*  Correspondence: cri_mini@yahoo.com

**Abstract:** A lush vegetation develops around the numerous wadi estuaries interrupting the Dhofari coast in Southern Oman. Many estuaries still house mangroves of *Avicennia marina* (Forssk.) Vierh., a very fragile ecosystem that is currently under threat in this area. A rather rich flora, strongly affected by the influence of the monsoon, grows in other estuaries. This study concerns the flora and vegetation of these peculiar habitats with special focus on the plants growing on the different substrates. To gain insights into the pollen diffusion and representation of these plants in the current pollen rain, we analyzed surface soil samples. By evaluating their pollen amounts, we related the modern pollen rain to the abundance/coverage of the plant species typical of the different environments. Rather than a punctual indication of the plants growing at a short distance, our pollen records offer a general picture of the flora and vegetation of the area. This information is crucial for the correct interpretation of pollen records from ancient soils, and underlines the utility of pollen analysis for the reconstruction of the vegetation history.

**Keywords:** flora and vegetation; modern pollen rain; pollen-vegetation relationship; pollen morphology; estuaries; mangroves; Dhofar

## 1. Introduction

Dhofar, the southernmost region of Sultanate of Oman (Figure 1), is the object of increasing interest both from a naturalistic and touristic point of view. Recent studies concern its geology, climate and the peculiar cloud forest, which covers the southern side of the mountain range, which runs parallel to the coast [1–4]. While the flora and vegetation of Dhofar have been the objects of study over a longer period [5–10], the knowledge of the pollen morphology and diffusion of the Dhofari plants is mostly lacking. An exception is the pollen belonging to the most common trees, shrubs and woody herbs [11,12], and some selected plants [13].

The attention devoted to the vegetation of Dhofar is born from the occurrence of plants adapted to a climate strongly influenced by the seasonal variation—under monsoonal climate—and the peculiar topography that make the region stand out from the rest of the territory of the Arabian Peninsula. The interest for this region also derives from the relationship that the local populations have developed with this land throughout its history. Dhofar, along with the rest of *Arabia felix*, has for centuries represented a crucial area for the commerce of frankincense and the site of prosperous ports such as the port city of Sumhuram, third century BC–fifth century AC [14,15], located in the estuary of Wadi Darbat (WD). The palaeoenvironmental [16] and archaeobotanical [17–19] studies in the area highlighted that the city flourished in a period characterized by a monsoonal circulation stronger than the present and a generally wetter climate. Progressively drier conditions and concurrent reduction in fresh water habitats coincided with the decline and consequent abandonment of the city [19].

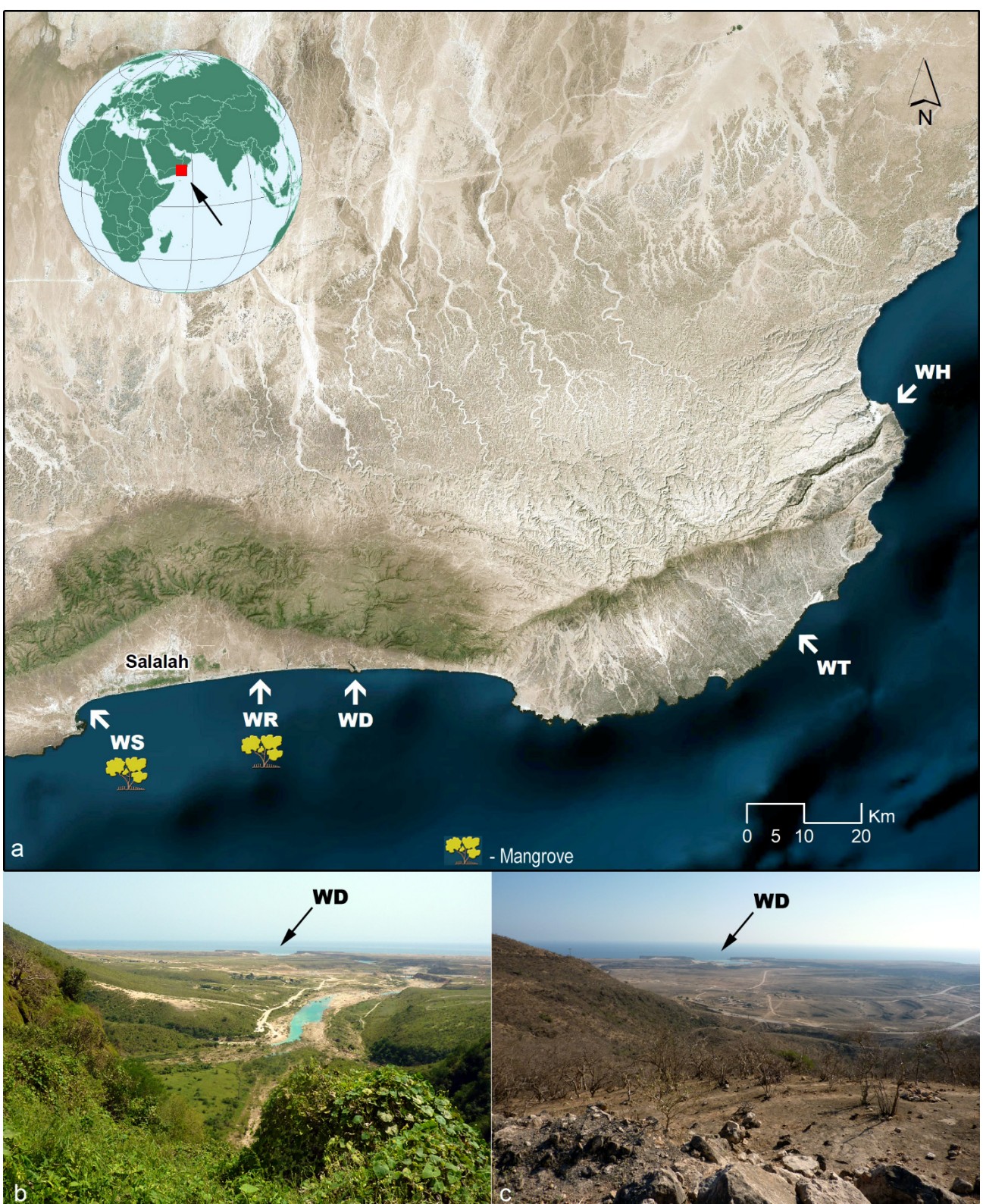

**Figure 1.** (**a**) The estuaries in exam along the Dhofari coast; (**b**) the Wadi Darbat course and Khor Rori (WD, arrow) from the Al-Qara Mountains at the end of summer; (**c**) the same view during the winter. WS—Khwar Qurm Al Sagir; WR—a khwar at East of Salalah; WD—Khor Rori; WT—the estuary of Wadi Ataq; WH—the estuary of Wadi Attabarran.

Another source of interest is the occurrence of peculiar habitats now threatened by the expansion of urban centers and construction of massive hotel structures, along with overgrazing and wood overuse and, in a broader context, by climate change [20]. At the end of the last century, Ghazanfar [21] had already raised awareness about the increasing threats to the plant diversity in Dhofar. More recently, attention has focused on the population decline of peculiar plants of the Dhofari flora, which has led to a more accurate study of their distribution, to the monitoring of the wild populations and, finally, to the planning of conservation programs [22,23]. Among them, special attention has been devoted to *Avicennia marina* (Forssk.) Vierh., a mangrove which grows at the estuary of seasonal watercourses (wadies). It was included as Least Concern in The IUCN Red List of Threatened Species [24] for the loss of its habitat. *A. marina* is now the object of a "Mangrove Transplanting Project in Oman" [25]. The beneficial role of these plant communities along the coasts is noteworthy and must be safeguarded: mangroves protect the shorelines from storms, limit the erosion of the coastline and sustain a high biological diversity. In addition, studies carried out in Northern Oman indicate *A. marina* as a plant potentially able to store and sequester high amounts of carbon [26].

Archaeobotanical studies have demonstrated that mangroves had a wider diffusion in the past by revealing the occurrence of both *Rhizophora* and *Avicennia* along the southeastern coast of Oman up to 6000 years before the present [27]. Therefore, *A. marina* communities also represent an important cultural heritage that requires safeguarding, just as much as the archeological sites do.

The mangroves of *Khwar ad Dahariz* in Salalah are a perfect example of a natural habitat that has been encroached upon by the city, where the mangroves have been cut and reduced to a bush (Figure 2), even if this estuary is part of the Khawrs Reserve of the Dhofar Coast and was still defined as being intact in 2004 [28].

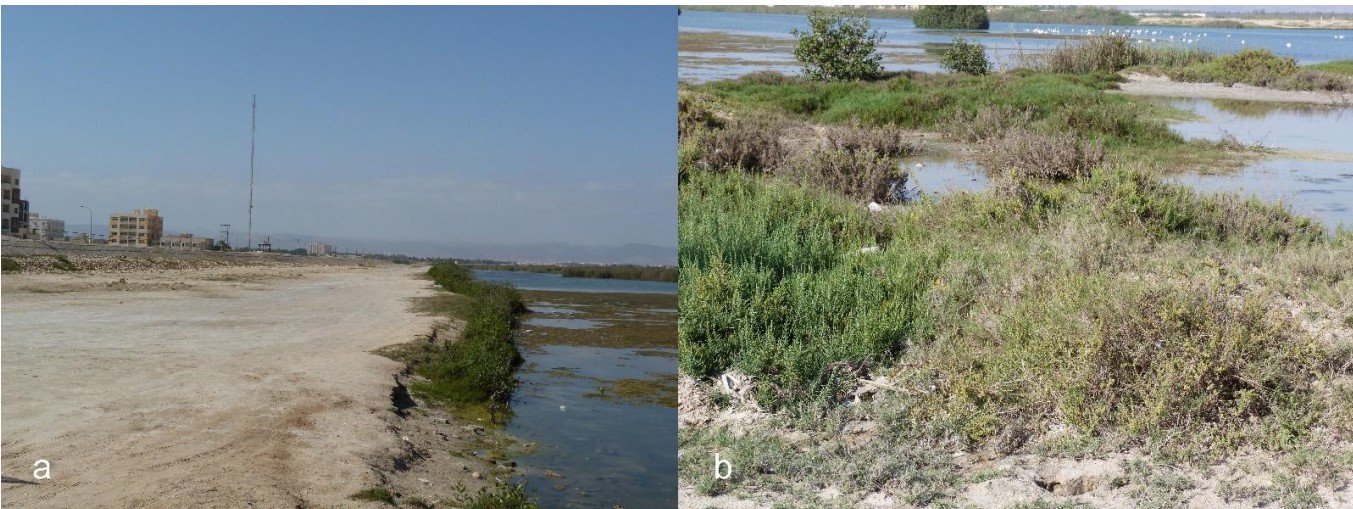

**Figure 2.** Khawr ad Dahārīz. (**a**) Degraded remains of *Avicennia marina* on the right bank of the estuary; (**b**) dominant *Arthrocnemum macrostachyum*, with *Suaeda vermiculata* and *Sporobolus spicatus*, on the sand bar closing the mouth of the estuary.

The aim of this study is to investigate the flora and vegetation that develop on the estuaries along the coast of Dhofar (Figure 1a) and the local pollen rain, with a special focus on the plant distribution and the pollen morphology of the main components. The palynological analysis of modern surface samples will allow us to gain insight into the pollen diffusion and representation of these plants in the current pollen rain. Indeed, the relationship between the pollen amount in the current pollen rain and the abundance/coverage of the plant species typical of the different environments will furnish crucial information for the correct interpretation of pollen records from ancient soils, which are the indispensable tool for reconstructing the vegetation history of these habitats.

To aid us in the identification of the grains in the surface samples, we studied the morphometric features of pollen grains of plants collected in the field. A brief description of the pollen morphology of the main species occurring in these environments is available in Supplementary S1.

## 2. Topography, Climate and Plant-Life of the Studied Area

The coastal area of Dhofar consists of a rocky plain facing the Arabian Sea that extends from the Yemeni border, at the southwest, to the Gulf of Hasik, at the northeast. It is interrupted by wadi estuaries (Figure 1a), which are barred at the mouth by a sand shoal during the dry season. The plain is flat in the surroundings of Salalah and characterized by flat apron fans at the east [4]. The Al-Qara Mountains (*Jibāl Al-Qārah*), an arched mountain range with a steep escarpment, run north of the Salalah plain (Figure 1a) and continue to the east reaching a height of 2000 m above sea level. These mountains form a natural northern border for the diffusion of the marine aerosol. Beyond the mountains, the vast desertic Arabian plateau extends to the north. The mountains also block the winds blowing along the Arabian Peninsula throughout the year and transporting large amounts of dust, so that they only weakly affect the coastal plain [29,30].

Regarding the climate, the western Dhofari coast falls under the Indian Ocean monsoon circulation. It extends from the Yemeni border to the Mirbat—Sadh peninsula, a Neoproterozoic outcrop between Mirbat and Hasik [31]. During summer (mid-June to August/mid-September), the south-western monsoon (Khareef) reaches speeds up to 15 m/s [32] and induces rainfall and dense fogs [3]. The average annual rainfall (1977–2003) is 112 mm in the coastal plain and 185 mm on the mountain escarpment [33]. Fogs, which may be considered a "horizontal precipitation", are usually not measured, but they are intercepted by plants, particularly by trees that are able to harvest the fog droplets dripping through their canopy [2]. A temperature inversion, caused by the warm northern winds, limits the inland movement of fog [34]. During winter, the northeastern dry monsoon develops over the Arabian Sea.

The concurrency of topography and climate make the Dhofari flora and vegetation a *unicum* in the Arabian Peninsula. Indeed, the southern slopes of the Jabal support a stable cloud forest, which can intercept a noticeable amount of water under its cover. Surprisingly, the forest seems to have undergone a slight increase in extension in the last few decades [1]. The forest is a drought-deciduous, broad-leaved plant community endemic of southern Arabia with the dominance of *Anogeissus dhofarica* A.J. Scott (≡ *Terminalia dhofarica* (A.J. Scott) Gere & Boatwr., Combretaceae) whose composition varies along the arch of the mountain range. This forest is described by the association Hybantho durae-Anogeissetum dhofaricae [35], which occurs from the Hawf Mountains in Yemen, through Jabal Qamr in Dhofar, up to the Al-Qara Mountains, further eastwards in Dhofar. The most typical variant of this forest grows on higher, southern slopes, best exposed to the southwest monsoon. In this variant, *A. dhofarica* is dominant, its canopy cover is the highest (60%), a shrub/lianas layer is also well developed, and the herb layer can reach 100% cover in the most favorable years. Moreover, a very rich epiphytic bryophyte flora develops on the tree trunks. Towards the western and eastern borders of its geographical range, as well as at lower elevations or on less favorable expositions, a more open forest, dominated by the tree *Blepharispermum hirtum* Oliv. (Asteraceae) occurs, down to an *Acacia-Commiphora* woodland where the exposure prevents the influence of the southwest monsoon. Growing on the plateau, where the moist gradient dramatically declines moving inland, is a wide grassland area, which according to Kürschner [35] is of anthropogenic origin. In natural conditions, the forest would also occupy the plateau, being then further inland and abruptly replaced by a xeromorphic dwarf scrub with dominant *Euphorbia balsamifera* subsp. *adenensis* (Deflers) P.R.O. Bally, typical of the surrounding south Arabian plateaus situated out of the southwest monsoon influence.

The lower part of the slopes, the foothills and the coastal plain also support lush vegetation during the Khareef, while they turn back to a desertic appearance in the winter

(Figure 1b,c). The foothills host xerophytic shrubs whose canopy, to a lesser extent than the forest at higher altitude, allow some dripping favoring the development of a rich undergrowth of herbs and grasses during the Khareef. The plain, on the other hand, is nowadays nearly deprived of any intercepting woody vegetation. As a result, the southwest monsoon sweeps the plain across without leading to any significant dripping phenomenon. Nonetheless, due to direct precipitations, which also occur during the Khareef, even the plain turns green (Figure 1b), through the rapid growth of ephemeral herbs and grasses [5]. It is assumed that the plain was formerly wooded. During World War II, the wood close to the foothills was cleared up for security purposes. Since then, further anthropic pressure, including bustle of off-road vehicles resulting in an impressive net of tracks, new road building, advancing urbanization and overgrazing, have dramatically reduced the extension and continuity of natural habitats in the plain [21].

As regards biogeography, the floristic composition of Dhofar can mainly be considered an extension of subtropical East African Flora, the "Somalia-Masai Region of endemism". Indicators of this predominant phytochorion are to be found in all natural habitats, including the coastal estuaries here surveyed [8].

## 3. Materials and Methods

The study concerns five estuaries (Figure 1a): WS—Khwar Qurm Al Sagir (Figure 3a,b); WR—a khwar at East of Salalah (Figure 3c,d); WD—Khor Rori, the estuary of Wady Darbat (Figure 1b,c and Figure 4a,b); WT—the estuary of Wadi Ataq (Figure 4c,d); WH—the estuary of Wadi Attabarran (Figure 4e,f). Two of them, WS and WR, host mangroves. These estuaries are named "khwar" precisely due to the mangrove occurrence. The estuaries not hosting mangroves are named "khor". The salinity of khwars varies from brackish inland to saline seawards, depending on the season, on local rainfall and groundwater seepage [8]. The estuaries currently devoid of mangroves have a lower salinity than those with mangroves [28].

WS (Figure 3a,b)—Khwar Qurm Al Sagir, 16°58′50.20″ N, 54°0′48.64″ E, is a short, ecologically constrained estuary in a coastal area heavily impacted by human activity. The estuary is surrounded by a wire fence and, to the northwest, by the main coastline road. Northeast of the estuary lies the adjacent lot of the Hilton Salalah Hotel. Southwest of the estuary, before the Raizut Harbour, a narrow, about 1.5 km long, more or less free coastal belt runs along the coastline road to the northwest, and the ocean to the southeast. The estuary is inhabited by mangroves that build a close curtain around the water. WS, together with WR, WD and WT, is part of a nature reserve since June 1997 (Royal Decree 49/1997), the Khawrs Reserve of the Dhofar Coast, which exists primarily for the protection of plants [21,36,37].

WR (Figure 3c,d)—Khwar al Qurm, east of Salalah, 17°1′43.02″ N, 54°17′13.74″ E, located in the Salalah plain near the Mosque Janoof, between wadi Razat and the Salalah Rotana Resort. WR is also a short estuary, but in a relatively less disturbed location than WS, notwithstanding an adjacent, vast touristic resort having its western border about 50 m east of the khwar. The nearest road is about 300 m north of the inland end of the estuary. With the exception of the enlarged southwest side facing the ocean, where the banks are nearly devoid of vegetation, a more or less continuous mangrove rim covers the wadi shores.

WD (Figure 4a,b)—Khor Rori, the estuary of Wadi Darbat, 17°1′53.58″ N, 54°26′13.81″ E. It is far larger and spectacular than the former two, and its existence is not under threat thanks to the presence of the important Sumhuram archaeological site, which rests on a hill between the terminal arms of the estuary. Nonetheless, extension and integrity of the surrounding natural area have been affected over the past decade by increasing disturbing factors, i.e., the building of a new main coastline road in 2018, which crosses the wadi upstream not far from the estuary and the rapid growth of the adjacent town of Taqah. With the exception of sparse trees, there is no woody vegetation along the estuary of Wadi Darbat.

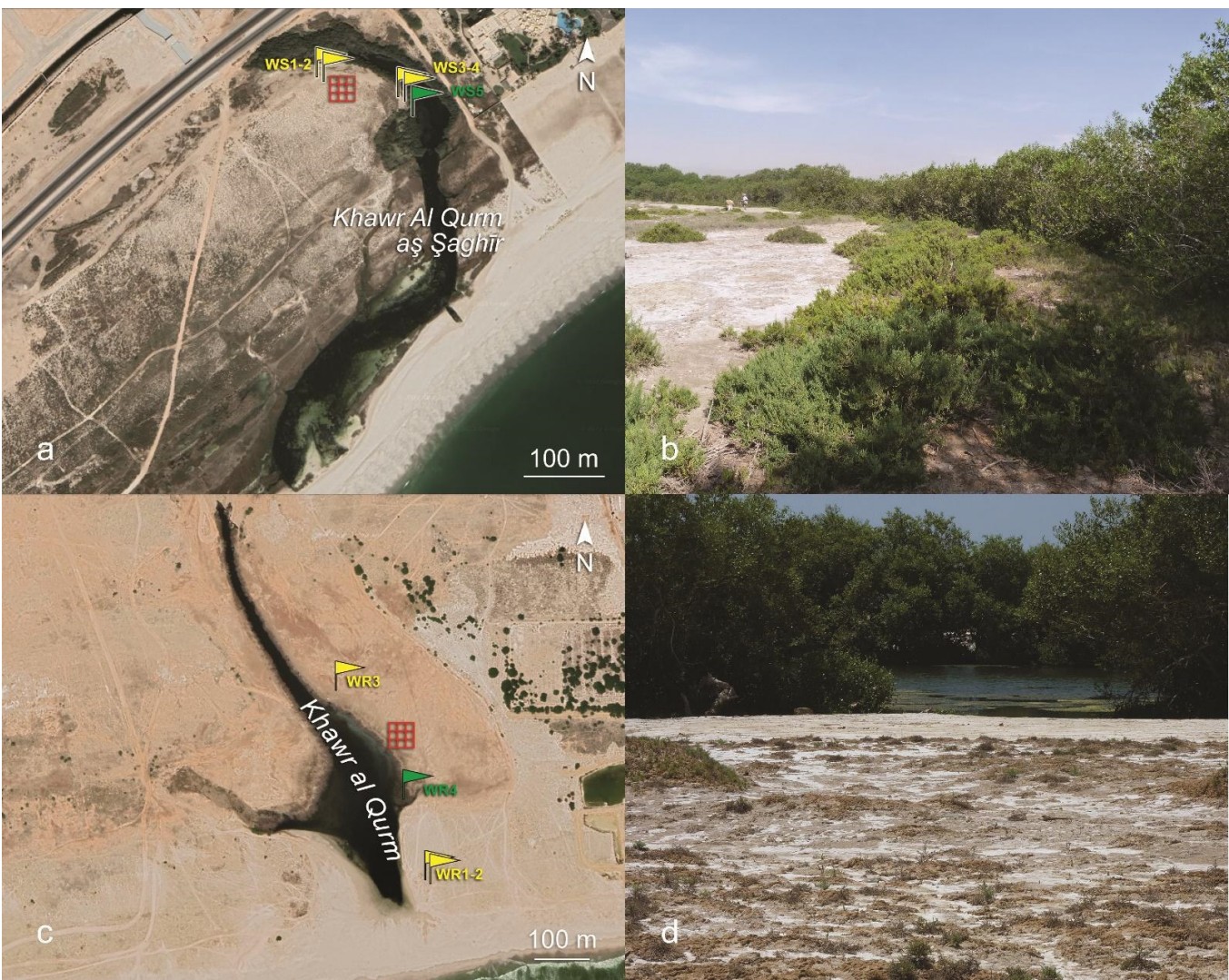

**Figure 3.** Estuaries with mangroves. Khawr Al Qurm as Saghīr (WS): (**a**) map; (**b**) the estuary with *Arthrocnemum macrostachyum* in foreground and *Avicennia marina* in the background. Khawr al Qurm (WR): (**c**) map; (**d**) the estuary with sabkha with *Cressa cretica* in foreground and *Avicennia marina* in the background. In the maps, the red grids indicate the location of the pollen samples and the flags the vegetation relevés. Green flag indicates mangrove, yellow flag indicates sand-sabkha vegetation.

WT (Figure 4c,d)—the estuary of Wadi Ataq, 17°5′39.85″ N, 55°7′20.48″ E, about 10 km east of Sadah. It is located along a rocky coast, represented by the southeastern slopes of Jebel Samhan, which is also a nature reserve, originally established (June 1997, by Royal Decree 48/1997) for the protection of remnants of deciduous tropical woodland and of possibly the last refuge of the Arabian leopard [21]. Jebel Samhan consists of limestone uncomfortably overlying a metamorphic basement of crystalline gneisses intruded by basalt dykes [38]. There is no coastal plain, except the relatively small area formed by the wadi deposits at the mouth of the estuary, surrounded by the outcropping gneiss basement with evident basalt dykes. Disturbing factors in this area are the coastline road, which crosses the wadi at a short distance upstream of the estuary, and animal grazing activities.

WH (Figure 4e,f)—the estuary of Wadi Attabarran, 17°26′41.64″ N, 55°15′4.26″ E, located in a moderately disturbed area east of the expanding new settlement of Hasik. It is amply out of the influence of the monsoon.

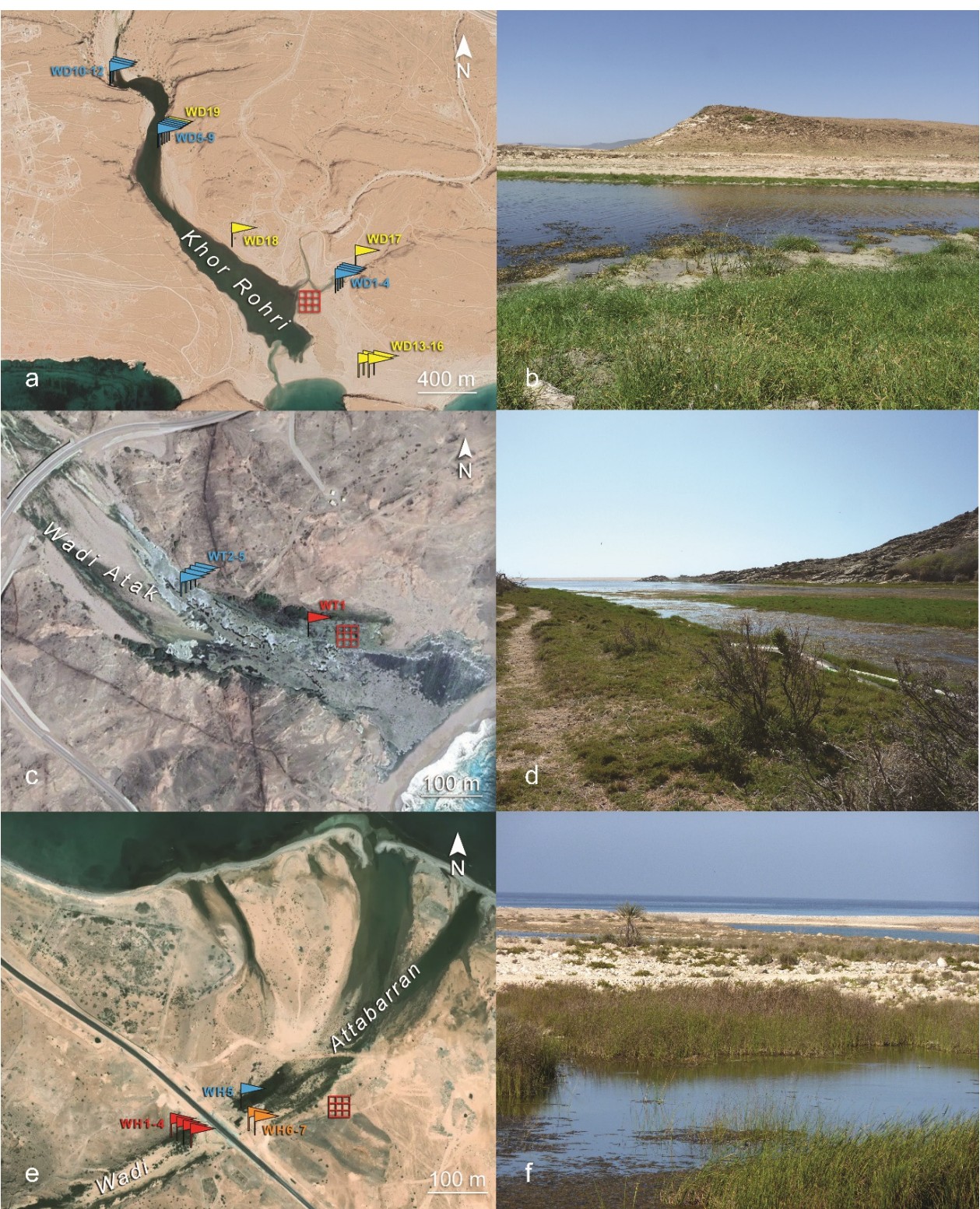

**Figure 4.** Estuaries devoid of mangroves. Khor Rohri (WD): (**a**) map; (**b**) wet wadi vegetation with *Cyperus laevigatus.* Wadi Atak (WT): (**c**) map; (**d**) *Indigofera oblongifolia* and dominant *Aeluropus lagopoides* in the foreground. Wadi Attabarran (WH): (**e**) map; (**f**) *Typha domingensis* in the foreground, with calcareous shore vegetation in the background. In the maps, the red grids indicate the location of the pollen samples and the flags the vegetation relevés. Yellow flag indicates sand-sabkah vegetation, blue flag wet wadi vegetation, red flag dry wadi vegetation, orange flag calcareous shore vegetation.

### 3.1. Floristic Composition of the Estuarine Plant Communities

In 2015, 36 vegetation surveys—here named relevés, French word for census—were carried out in the immediate vicinity of the pollen rain grids (see below, Figure 3a,c and Figure 4a,c,e) at the end of the Khareef season (end September, beginning October), when Dhofar plant diversity shows its highest values of the year. The object of these relevés were *phytocoenoses*; real vegetation units subjectively (not randomly) identified by floristic and ecological homogeneity. Within each phytocoenosis habitus, floristic composition of the plant community and topographical characteristics are expected to be homogeneous, with heterogeneity reduced as much possible [39]. Therefore, in order to survey the surroundings of each pollen rain grid, a variable number of relevés was carried out, depending on site heterogeneity. The size of the single relevé depends on the phytocoenosis texture, being at least as large as to include all the *taxa* occurring in each phytocoenosis. Within each relevé, geographical coordinates, vegetation total cover, surface area, orientation, inclination, altitude above sea level, census of plant species and their specific cover were recorded. Species cover was taken according to the Braun–Blanquet [40,41] species cover scale, which ranges from 1 to 5 (corresponding, respectively, to the cover percentage intervals <1%, 1–5%, 5–25%, 25–50%, 50–75%, 75–100%). Furthermore, the addition of a "+", was generally used for abundant ephemerals with low cover (e.g., grasses or slender herbs) and "r" (rare) for ephemerals present with 1(−2) individuals. In this survey, "+" and "r" were sometimes used for sparse, low covering seedlings of perennials too. Beside each Braun–Blanquet cover value, we also reported the cover percentages for each species. Since only two of the surveyed plant communities presented a certain—very low—degree of stratification, no stratified relevés were carried out. It follows that in these two relevés (WR2 and WT1) the sum of the single plants coverage is slightly higher than the total coverage. The identification of the *taxa* was made mainly by means of Flora of the Sultanate of Oman [9,10,42,43] and Flora of the Arabian Peninsula and Socotra [44,45]. Nomenclature was then crosschecked and updated according to POWO [46].

#### 3.1.1. WS

Our survey (Figure 3a) includes two relevés upon and at the edge of a low sand dune with sparse limestone outcrops about 10 m west of the estuary, and two relevés at closer distance from the estuary on hardened sand/loam, salty substrate, all on grass/herb vegetation. A fifth relevé was taken within the mangrove belt.

#### 3.1.2. WR

The survey (Figure 3c) includes two relevés on the ocean side, a few meters from the south end of the estuary, on loose sandy substrate. The first of these relevés was made on a sparse sedge vegetation, the second one beneath the thick canopy of an isolated legume tree. A third relevé was taken about 350 m north-northeast of the former, just outside of the mangrove rim, on hardened sand/loam, salty substrate with low herbs. A fourth relevé was taken within the mangrove belt.

#### 3.1.3. WD

The riverbank vegetation was surveyed at three locations placed along the border of the wadi at decreasing distances from the estuary mouth. In each location, relevés (Figure 4a) were made along a moisture gradient, from the submerged bank of the river up to the emerged wet meadows in muddy soils. The low herbaceous, sparse vegetation of the loose, sandy and more or less hardened loamy, salty soils was also surveyed, in four locations at a higher distance from the estuary branches.

#### 3.1.4. WT

Relevés (Figure 4c) were taken on one of two dry strips of land running down to the sand bar separating the khwar from the ocean. The left strip, adjacent to the flooded, marine end of the estuary, was chosen due to its width and thriving vegetation. Further relevés

were taken upstream within the drained wadi bed, following the increasing moisture gradient occurring from the wadi bank to the central axis of the wadi.

### 3.1.5. WH

The relevés (Figure 4e) were carried out in a dry stretch of the pebbly wadi bed upstream of the coastline road, on a massive calcareous shore outcropping downstream of the coastline road, and in a flooded depression near the mouth of the estuary.

### 3.2. Current Pollen Rain and Pollen Morphology

To study the current pollen rain we analyzed the pollen content of surface soil samples collected in proximity of the relevés (Figure 3a,c and Figure 4a,c,e). To avoid the proximity effect of the plants, more samples (5–9) were collected at each site: as far as possible, a virtual square grid was traced on a homogeneous area of the estuary bank, taking care to avoid the areas subjected to trampling. The side of the grid was ca. 30 m, and each square measured 10 m. Up to nine samples were collected at the vertices of the grid squares, excluding water-soaked soil and sand. For evaluating the amount of airborne pollen, the samples of each grid were treated as subsamples and the average pollen percentages and the average pollen amount were calculated and related to the abundance/coverage of the plants. Therefore, each site is represented by one pollen sample.

The soil samples were treated according to the routine methodology, including HCl, HF, NaOH treatments, and the acetolytic method [47]. Absolute pollen frequency (APF) is calculated as the number of grains per gram of known sediment weight.

Pollen identification was made with the help of the literature [11,48] and the observation of the pollen morphology of the reference material purposely collected during our missions in Dhofar. Flowers of the most common plants of each site were directly collected during the field activity. When no plant was blooming, flowers/anthers were sampled from exsiccata collected during the previous visits to Dhofar and deposited in the Centro Studi Erbario Tropicale of Florence (FT). The pollen grains were analyzed under a light microscope (LM) and a scanning electron microscope (SEM). For LM analysis, pollen grains were acetolyzed [47] and included in a 50% water/glycerol solution *v/v*. For SEM analysis, acetolyzed pollen grains were dehydrated through a series of ethanol solutions of increasing alcoholic concentration, critical point dried, and then gold-coated. All the measurements were performed at LM, at a magnification of 630x, on at least 30 grains per sample, randomly selected. The measurements included the length of the polar (P) and equatorial (E) axis or the diameter (D; mean diameter or mean length of the apertures; exine thickness measured in the mesocolpium. Pollen shape classes (P/E ratio) are according to Erdtman [49]. Terminology is according to Punt et al. [50]. (Supplementary S1). The morphometric features of the pollen grains belonging to species of the Amaranthaceae family were the object of a detailed examination. The observation took into account: the diameter, the thickness of the exine, the pores' diameter, and the distance between them. These data have been used to carry out a statistical-comparative analysis with the aim of finding a correspondence between the grain morphology of these plants. The study has been developed using the FD package [51] with R-Studio software [52]: the package allows us to graphically and quantitatively examine the functional traits between two or more species that are divided in a cluster dendrogram resulting from a similarities/dissimilarities coefficient reported on a scale. The analysis considered the average values between the morphological features of the pollen grains (Supplementary S2).

The pollen diagram was drawn using TILIA 2.0 [53] and shows the average pollen percentages of the plants recorded in the surveys. In order to compare the current pollen rain to the plant abundance/coverage percentages, the pollen *taxa* are grouped according to the different types of habitats delineated by the vegetation surveys. The *taxa* (from species to family) occurring in more than one habitat are repeated in the diagram.

## 4. Results

### 4.1. Relevés

#### 4.1.1. WS

(Supplementary S3 Table S1)—Coast sand-sabkha vegetation (WS1–4). WS1. The phytocoenosis occupies the vast plateau-like top of a shallow sand rise and it consists of sparse, firm tufts of *Urochondra setulosa* (Trin.) C.E. Hubb. neatly emerging from the sand bed. *Urochondra setulosa* is a turfy grass typical of sandy sabkhas and saline flats along the whole Oman coast [44]. In the surveyed sites, it was mainly observed on loose sand rises and low dunes. WS2. At the border of the sand rise, where the substrate is somewhat intermediate between loose sand and hardened loam, a slight slope hosts a more diversified plant association. It includes grasses with dominant *Sporobolus spicatus* (Vahl) Kunth, followed by Urochondra setulosa and *Aeluropus lagopoides* (L.) Thwaites. The dominance of *Sporobolus spicatus* and the presence of *Aeluropus lagopoides*, as well as that of the Amaranthaceae *Sevada schimperi* Moq. and *Suaeda aegyptiaca* (Hasselq.) Zoh., may reflect a higher silty fraction and increased salinity than on top of the dune. WS3–4. Close to the abruptly arising mangrove stand, on a hardened silty/sandy saline substrate, alternating patches of *Sporobolus spicatus* and *Arthrocnemum macrostachyum* (Moris) K. Koch occur (WS3), with the latter occasionally being dominant (WS4).

Mangrove (WS5). The mangrove is a nearly closed, monospecific grove of *Avicennia marina* (Forssk.) Vierh., covering approximately 95% of the soil surface.

#### 4.1.2. WR

(Supplementary S3 Table S2)—Coast sand-sabkha vegetation (WR1–3). WR1. The loose sand of the beach dune opposite the khwar and facing the ocean hosts spare vegetation (total cover 5%) dominated by the common tufted sand sedge *Cyperus conglomeratus* Rottb. and accompanied by *Sporobolus spicatus* and *Aeluropus lagopoides*, the halophyte Amaranthaceae *Suaeda vermiculata* Forssk. ex J.F. Gmel. and the more ubiquitous small Aizoaceae *Aizoon canariense* L. WR2. This phytocoenosis is identified by the canopy outline of an alien legume tree/shrub, *Prosopis juliflora* (Sw.) DC., by now a widespread and worrisome weed in Oman, after its introduction as a landscape tree [43]. In the Dhofar coastal plain, the species is intruding upon depleted, autochthonous vegetation. No undergrowth is present under the thick canopy (100% cover), except sparse individuals of *Sporobolus spicatus*, and elsewhere, the relatively dominant sand specialist *Cyperus conglomeratus* is absent here. WR3. The silty, saline flat beside the khwar hosts vast, open populations of the small halophytic Convolvulaceae herb *Cressa cretica* L. (Figure 3d).

Mangrove. WR4. As in WS, the mangrove is a close monospecific population of *Avicennia marina*.

#### 4.1.3. WD

(Supplementary S3 Table S3a,b)—Wet wadi vegetation (WD1–12). WD1–3. Due to the extent and complexity of the Wadi Darbat estuary, with respect to the other estuaries on the plain, the wet vegetation was investigated on several of its arms, beginning from the submerged margin up to the emerged, muddy banks. One series of relevés was taken in a narrow secondary left arm, east of the ancient Sumhuram. The central part of the arm (WD3) was scarcely colonized by any emerging vascular plants, but richly inhabited by the submerged fresh and brackish water hydrophyte *Najas marina* L. (100% cover). The submarginal, shallow water of the arm (WD1) was colonized by young, loose settlements of the perennial helophyte sedge *Schoenoplectus subulatus* (Vahl) Lye, which is relatively dominant on the tropical-subtropical weed *Paspalum vaginatum* Sw., a creeping species thriving in wet habitats with brackish water, and on sparse *Najas marina*. The partly emerged, partly shallowly inundated margin (WD2) was dominated (50% cover) by the tropical sedge *Cyperus laevigatus* L. (Figure 4b), typical of moist saline and brackish mud, interspersed with *Paspalum vaginatum* and sparse *Sporobolus virginicus* (L.) Kunth, a pan-tropical grass of sand dunes and brackish swamps. WD4. An adjacent, seaward segment

of the same estuary arm, submerged and bordered by outcropping rocks, bore a different plant ensemble, i.e., a dense population of the tall rhizomatous reed *Phragmites australis* (Cav.) Trin. ex Steud. subsp. *altissimus* (Benth.) D. Rivera & M.A. Carreras, with a minor percentage of the evergreen, scrambling shrub *Salvadora persica* L. Both species are common in saline to brackish swamps and soil.

WD5–9. The left bank of the main estuary, upstream of the ancient city, being larger and deeper than the secondary branch, was colonized by a plant community differing in age and habit. The submerged margin of the estuary (WD5) was colonized by *Najas marina* and *Schoenoplectus subulatus*, the former showing the higher coverage (75%), the latter forming large dark green tufts rising up about 1 m above water level and covering 25% of the water surface. The coverage within each tuft of *Schoenoplectus subulatus* was more or less 100% (WD6). The barely emerging bank belt (WD7) was occupied by a continuous (100% cover) wet meadow of codominant *Cyperus laevigatus* and the Scrophulariaceae *Bacopa monnieri* (L.) Wettst. The latter is a pantropical, creeping herb of permanent pond and stream margins. Further up the emerged bank, *Cyperus laevigatus* became dominant, with scanty *Bacopa monnieri* and *Schoenoplectus subulatus* (WD8), followed by a wet meadow belt of *Paspalum vaginatum*, with minor *Bacopa monnieri*.

Upstream of WD5–9, on the right bank of the estuary, a bight was surveyed, where three phytocoenoses (WD10–12) were recognized: a closed (100% cover) monospecific, helophytic *Schoenoplectus subulatus* community (WD12) in the submerged core of the bight; a likewise closed community with coexistent *Phragmites australis* subsp. *altissimus* and *Typha domingensis* Pers. (the latter characteristic of permanent, fresh or brackish water pools) in a submarginal position; an adjacent wet meadow with 90% total cover dominated by *Paspalum vaginatum* with subordinate *Cyperus laevigatus*, *Bacopa monnieri*, and sparse *Sporobolus spicatus*.

Coast sand-sabkha vegetation (WD13–19). WD13–16. Phytocoenoses of loose sand and sabkha (hardened sandy-silty saline flats) were investigated on the hydrographic left side of the estuary, east of the sand bar closing the estuary from the ocean. WD13 was taken in a plain with scanty vegetation (total cover 20%) of raised *Urochondra setulosa* mounds surrounded by shallow ditches with *Cyperus conglomeratus* and other underrepresented species like *Sporobolus spicatus*, *Tetraena simplex* (L.) Beier & Thulin, *Heliotropium bacciferum* Forssk. *Cleome brachycarpa* Vahl ex DC. and *Aizoon canariense*. Quite similarly, WD16 was characterized by *Urochondra* mounds surrounded by saline shallow ditches with *Sporobolus spicatus* and sparse *Cyperus conglomeratus*. WD14, made seawards on loose sand showed a more diverse community dominated by the palaeotropical grass *Sporobolus ioclados* (Nees ex Trin.) Nees (a species with a broad tolerance range, which can be found in rocky slopes, seashores and mangrove swamps in Oman) with *Sevada schimperi* and *Polycarpaea spicata* Wight ex Arn., besides the same species present in WD13. WD15 was carried out on a hardened, saline flat and included as co-dominant the sabkha characteristic species *Cressa cretica* and *Sporobolus spicatus*. Exactly the same phytocoenosis was observed in WD17, at the landward end of the left arm of the estuary, near WD1–3. A third variant, combining sand and sabkha communities, was observed in WD18, SW of Sumhuram, where the *Urochondra* raised tufts were surrounded by saline shallow ditches with codominant *Cressa cretica* and *Sporobolus spicatus*. *Sporobolus spicatus* was also observed in monospecific populations (WD19) with high coverage (80%), on sand at the external side of the riverine phytocoenoses WD5–9, moving up from the estuary bank.

### 4.1.4. WT

(Supplementary S3 Table S4)—Marginal dry wadi vegetation (WT1). In this estuary, stands of *Salvadora persica*, *Tamarix* sp. and *Phragmites australis* occur on the margin of the alluvial fan, within and around the seasonally flooded area. *Salvadora persica*, particularly abundant, is not restricted to the estuary deposits, but spreads out on the surrounding basement outcrops, being a showy element of the landscape enclosing the estuary. Abundant among the *Salvadora* shrubs was also the herbaceous Malvaceae *Senra incana* Cav.,

a common weed of waste sites, but also of wadi beds. Inside this ecotonal, discontinuous belt, the dry land strip parallel to the flooded, marine end of the estuary was surveyed (WT1). It was a drained meadow with shrubs, with 90% total coverage. The shrubs were represented by the wadi and coastal plain legume *Indigofera oblongifolia* Forssk. (Figure 4d), with a low degree of coverage (about 15%) apparently due to foliage impoverishment caused by overgrazing. The underlying meadow was dominated by *Sporobolus virginicus* with lower *Aeluropus lagopoides*. Both species are characteristic of saline, brackish habitats with varying degrees of drainage. Individuals of *Cressa cretica* were also observed.

Wet wadi vegetation (WH2–5). Upstream along the estuary, a series of relevés was conducted where the estuary was not submerged, but was still wet, from the border to the center of the estuary. At the wadi bed margin, outside the alluvial wadi substrate—i.e., upon the crystalline basement—a thick meadow (100% cover) dominated by *Sporobolus virginicus* and, secondarily, by the swamp sedges *Cyperus rotundus* L. and *Eleocharis geniculata* (L.) Roem. & Schult. occupied the left bank (WT2). Adjacent to WT2, toward the center of the wadi, the dominance of *Sporobolus virginicus* fell, and *Sporobolus spicatus* took its place, with *Sporobolus virginicus* and *Cyperus rotundus* at lower percentages (WT3). Further down into the wet wadi centre, *Eleocharis geniculata* became dominant, still accompanied by *Sporobolus virginicus* and sporadic *Cyperus rotundus* (WD4) with the total cover slightly dropping. The center of the wadi, still partly flooded, was dominated—with the total cover dropping further—by *Bacopa monnieri*, accompanied by *Sporobolus virginicus* and *Eleocharis geniculata* (WT5). Notably, *Typha* sp., which had been observed before 2015 in this estuary, by September 2015 had disappeared and therefore is not represented in the relevés.

4.1.5. WH

(Supplementary S3 Table S5)—Marginal dry wadi vegetation (WH1–4). The chosen dry segment of the wadi bed showed a left raised embankment of gravel/pebble with thick, 3–4 m high vegetation (100% total cover). Codominant were *Phoenix dactylifera* L., *Phragmites australis* subsp. *altissimus*, *Tripidium ravennae* (L.) H. Scholz (a species growing along watercourses) and *Sporobolus virginicus*, less important coverage was shown by *Tamarix mascatensis* Bunge (species of wadi beds, sand and saline soils) and *Indigofera oblongifolia*. The climbing perennial Apocynaceae herb *Pentatropis nivalis* (J.F. Gmel.) D.V. Field & J.R.I. Wood, often appearing in wadi beds had low cover values, but a diffused presence with its over scrambling stems (WH1). Moving down into the wadi bed were thin woodland (45% total cover, 3 m high) with codominant *Tamarix mascatensis*, *Indigofera oblongifolia*, the grass *Sporobolus virginicus* and the creeping Verbenaceae *Phyla nodiflora* (L.) Greene took place (WH2), followed further in the wadi bed by even thinner woodland (40% cover, 2–3 m high) with the coastal plain legume tree *Vachellia tortilis* (Forssk.) Galasso & Banfi and *Tamarix mascatensis*, with few bushes of *Tephrosia purpurea* (L.) Pers. subsp. *apollinea* (Delile) Hosny & El-Karemy, a subspecies thriving in gravelly wadi beds (WH2). The center of the dry gravelly/pebbly wadi bed harbored a sparse shrubland (35% cover, 1–1.5 m high), which beside *Tamarix mascatensis*, minor *Indigofera oblongifolia* and *Tephrosia purpurea* subsp. *apollinea* included the aromatic composite *Pluchea arabica* (Boiss.) Qaiser & Lack (indicator of high groundwater level) and individuals of the more ubiquitous, but common in pebbly coasts *Heliotropium bacciferum* Forssk.

Wet wadi vegetation (WH5). Downstream, near the mouth to the ocean, the estuary was flooded and colonized by a thick, monospecific population of the helophyte *Typha domingensis* (100% cover, Figure 4f).

Low, flat calcareous (madreporic) shore vegetation (WH6–7). To the side of the estuary axis, on the white calcareous shore, two slightly different habitats were surveyed; their difference depending on the level of degradation in the rocky substrate. On a bare outcrop without debris, the vegetation was limited to cracks and interstices between blocks and showed a particularly low cover (15%). The most abundant species was *Pluchea arabica*, followed by the desert and pre-desert composite *Pulicaria glutinosa* (Boiss.) Jaub. & Spach (common in wadi beds as well), by *Salvadora persica* and the Amaranthaceae *Wadithamnus*

*artemisioides* subsp. *batharitica* (A.G. Mill. & J.A. Davis) T. Hammer & R.W. Davis (typical of limestone cliffs and rocks), plus some other scanty presences (WH6). Where the shore was partly degraded to gravelly debris, the vegetation was slightly richer in cover (20%) and diversity. *Indigofera oblongifolia* and *Pulicaria glutinosa* dominated, followed by *Salvadora persica*, the chasmophile *Lavandula macra* Baker (Lamiaceae) and *Petroana montana* (Balf. f.) Madhani & Zarre (Caryophyllaceae), and many other species characteristic of gravelly wadi beds, like the regional endemic *Polygala mascatensis* Boiss. (Polygalaceae) (WH7).

### 4.2. Current Pollen Rain

The superficial soil sediments from the estuaries presented a considerable amount of pollen, although many grains were folded or broken. The APF is remarkably variable, with mean values from 3900 (WT) to 10,400 grains/gram (WS) in the khwars, and from 3500 to 5600 in the khors.

The samples show the great diffusion of the pollen of Amaranthaceae and Poaceae all along the coast of Dhofar. Amaranthaceae are especially abundant in khwar WS and WR, Poaceae are everywhere present and produce large amounts of pollen grains that, unfortunately, are not identifiable at the genus level. Our attempt at identifying Amaranthaceae pollen (Supplementary S1) at the genus level was not entirely successful. The similarity of the analysis conducted on the morphological features of the acetolysed pollen grains of the Amaranthaceae species only allows us to divide them into two groups (Supplementary S2). Unfortunately, the state of preservation did not permit recording the morphometric features of a noteworthy part (about 30%) of the grains making the accurate measurements of the well-preserved grains mostly unsatisfactory.

In the pollen records, the pollen grains ascribable to the plants recorded in the relevés range from 51.8% (WT) to 85.6% (WS) of the total grains and are below 30% if we exclude Poaceae and Amaranthaceae (Figure 5).

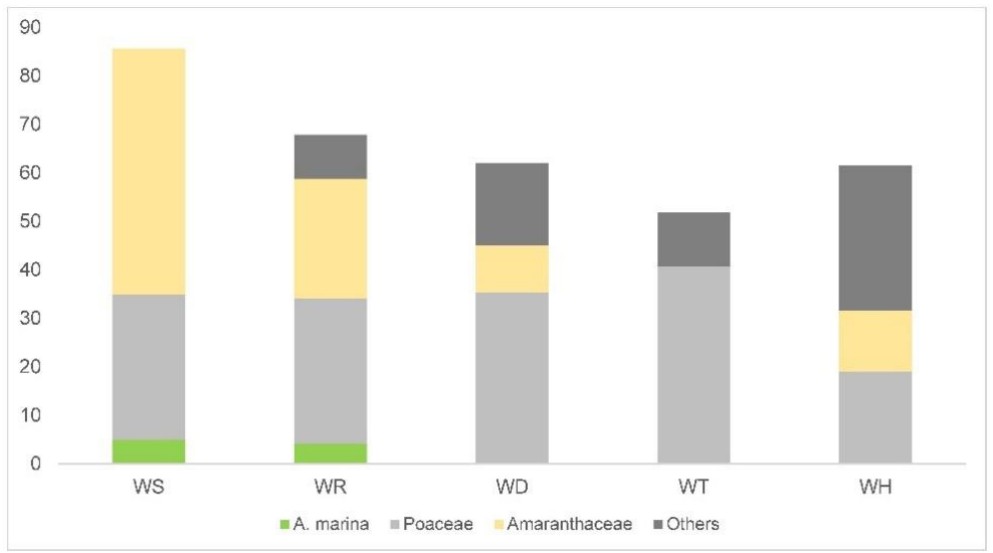

**Figure 5.** Pollen percentages of the main plants recorded in the relevés.

At WS and WR, the two estuaries where mangroves occur, *Avicennia* pollen was recorded at 5% and 4%, respectively. It is important to stress that the values in the diagram (Figure 6) are the average pollen percentages of subsamples collected in virtual grids (see Materials and Methods Section 3.2). Considering the subsamples separately, the values are very different in the soil sediments collected near *Avicennia* that reach 13.6% in WS and 24.6% in WR, from the furthest ones to *Avicennia*, only 0.5% in WS and 0.6% in WR.

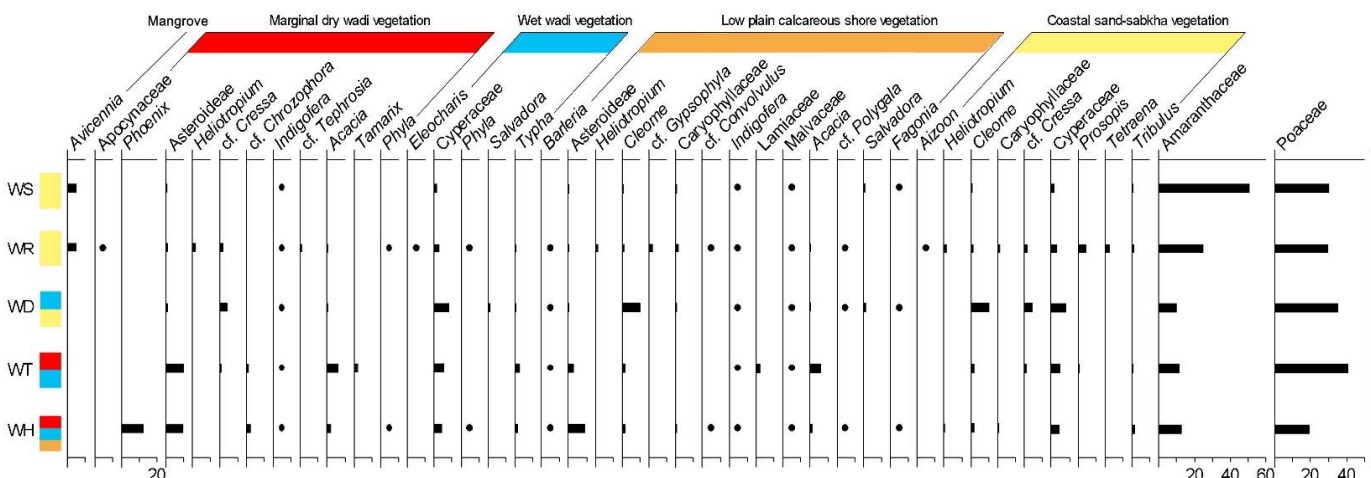

**Figure 6.** Pollen diagram showing the average pollen percentages (bars) of the surface samples collected in the five estuaries. Selected *taxa*. ● = Presence < 1%. Red = Marginal dry wadis; Blue = Wet wadis; Orange = Low plain calcareous shores; Yellow = Coastal sand-sabkhas.

In both these estuaries, but especially in WR, all of the coastal sand-sabkha plants are well-represented, e.g., *Prosopis* that here reaches its highest value of 4%.

Pollen from plants belonging to Wet wadi vegetation are especially represented by Cyperaceae, which also occur in the coastal sand-sabkhas vegetation. These plants are scarcely represented in the khwars WS and WR and more abundant in the khors WD, WT and WH.

Noteworthy in WD is the occurrence of pollen of plants from the coastal sand-sabkha vegetation, overall *Cleome* and *Cressa* that only here show values of 10% and 4%, respectively. *Acacia* shows its peak of 6% in WT, along with *Tamarix* (2%), both members of the marginal dry wadi habitat. Furthermore, from the marginal dry wadi vegetation is cf. *Chrozophora* that in WH reaches 2%, while the noteworthy occurrence of Asteroideae pollen (9%), can be connected to plants belonging to the low plain calcareous shore vegetation. WH is also the only site where *Phoenix* was recorded with the significant percentage of 12% (Figure 6).

## 5. Discussion

Our study allowed us to evaluate the pollen diffusion of the various plants growing along the wadi estuaries, both khwars (mangrove sites) and khors (non-mangrove sites), along the Dhofari coast and formulate some more general considerations on the relationship between vegetation and pollen rain, by comparing the list of plants occurring in the estuary vegetation with the list of pollen *taxa* in the surface soil samples. The study of this relationship is a crucial point of reference for the interpretation of ancient pollen records used in the reconstruction of the vegetation history of an area and to gain insights on the climate and the diet of the past inhabitants [19,54].

The vegetation survey in the khwars WS and WR pointed to low species diversity in this habitat, the monospecific mangrove being closely surrounded by a scanty greenery belt represented by salt and disturbance tolerant coastal sand-sabkha vegetation dominated by Amaranthaceae and Poaceae and totally lacking the plants of the wet wadi vegetation. These two concurrent conditions are connected to the high salinity levels of the soils surrounding the khwars that vary from brackish inland to saline seawards [8], while the khors have a lower salinity [28] and host more varied vegetation. The size of both mangrove sites WS and WR is small and their survival in the long run is quite uncertain, unfortunately. This situation highlights the importance of conservation projects focused on these particular habitats. Nonetheless, existing valiant projects [28] involving protective measures on one hand and active measures like mangrove afforestation on the other, have until now hardly been fulfilled in the Dhofar plain that over the past ten years has seen rampant and uncontrolled urban expansion.

Although the vegetation survey was carried out in the season of highest plant activity, ephemeral and emicryptophytes that do not vegetate and flower in the Khareef season were not represented in the relevés, whereas they were detected in the pollen analyses. In the same way, some differences between pollen grids and relevés may be connected to the fluctuations of monsoon strength from one year to the other, since the two sampling strategies were mostly carried out in different years and the sediments contain pollen accumulated over the years.

The surface soil samples show the great diffusion in the pollen of Amaranthaceae and Poaceae all along the coast of Dhofar. Amaranthaceae pollen is abundant in all of the records, even when they are not growing in proximity to the sampling site, like in WT (11.4%, Figure 6). However, the highest values of Amaranthaceae pollen are still recorded where these plants are dominant in the vegetation. With the exclusion of Amaranthaceae and Poaceae, the pollen of all the surveyed plants is below one-third of the pollen rain content. Since we cannot affirm with certainty that their pollen belongs to the actual surveyed plants, as it is more likely coming from plants outside the relevés, the contribution of the local plants could be even lower.

By examining the complete list of plants surveyed in the study (Supplementary S3 Tables S1–S5), we can evince that some of them only appear in the pollen records when they also occur in the relevés, i.e., in the immediate vicinity of the surface sampling points (Figure 5, Table 1). A good example of a discriminatory high-fidelity species is *Avicennia marina*, known for its low pollen dispersal [12], as also shown by a modern pollen rain study carried out in an estuary in New Zealand [55]. *Aizoon* and *Cressa*, small entomogamous weeds, are other good examples of discriminatory *taxa*. We can suppose, therefore, that the pollen of these plants is a reliable indicator of their local presence and of the characteristics of the surrounding vegetation. However, a larger number of sampling sites than the one presented here is necessary for a quantitative assessment of the representation of the pollen *taxa* through fidelity indices [56]. *Phoenix*, on the other hand, represents a special case in light of its peculiar artificial pollination mechanism and domestication history [12,54].

For the reconstruction of the vegetation history of Dhofar, attention was often focused on the presence vs. absence of mangroves. In the case of WD, for example, a hypothesis was that mangroves were growing in the estuary during the settlement of the city of Sumhuram [57]. The palynological analyses of a core from Khor Rori and several stratigraphies in the city [16,18] have not confirmed this scenario. Our data indicate that mangroves in the area, at least *A. marina*, have poor pollen diffusion [12], but a more targeted sampling might be necessary to definitely answer the question of its past presence.

Other plants appear in the pollen records both when they occur in the relevés and when they do not (Figure 5, Table 1). These plants are widespread along the coast of Dhofar and in the inland region. An example is the Asteraceae family, which have herbaceous members as well as numerous arboreal species like *Blepharispermum hirtum*, that grow in the cloud forest on the inland slopes of the Al-Qara Mountains and *Prosopis*, with scattered individuals occurring throughout the plains behind the coast. As expected, however, the pollen percentages of these plants are always higher when they also occur in the sampling area. Other plants whose pollen frequently appears in the records (Figure 6) are *Indigofera*, Malvaceae (cf. *Senra*) and *Tribulus*, all occurring with scattered individuals along the coast and on the inland plateau. *Tribulus*, in particular, thrives on terrain heavily impacted by trampling.

Finally, other plants, like *Najas* and *Bacopa*, never appear in the pollen records, even when they occur in the relevés (Supplementary S3 Tables S3 and S4). *Najas* is an aquatic hydrogamous plant characterized by pollen grains with very thin exine structure [58,59] that consequently breaks down easily during the chemical treatment of pollen samples. *Bacopa* is a tiny plant growing in close proximity to water or in calm shallow waters and has entomogamous pollination mechanisms (Table 1).

**Table 1.** Comparison between plants in the relevés and in the pollen rain.

| Plants in the Relevés | Pollination Syndrome | Plants in the Pollen Rain | | | | |
|---|---|---|---|---|---|---|
| | | **WS** | **WR** | **WD** | **WT** | **WH** |
| *Avicennia* | E | + | + | − | − | − |
| *Barleria* | E | − | + | + | + | + |
| *Aizoon* | E | + | − | − | − | − |
| Amaranthaceae | A,E | + | + | + | + | + |
| Apocynaceae | E | − | + | − | − | − |
| *Phoenix* | * | − | − | − | − | + |
| Asteroideae | E | + | + | + | + | + |
| *Heliotropium* | E | − | + | − | − | − |
| *Cleome* | E | + | + | + | + | + |
| Caryophyllaceae | E | − | + | − | − | + |
| Convolvulus | E | − | + | − | − | + |
| *Cressa* | E | − | + | + | + | − |
| Cyperaceae | A | + | + | + | + | + |
| Chrozophora | E | − | − | − | + | + |
| *Najas* | H | − | − | − | − | − |
| Lamiaceae | E | − | − | − | + | − |
| *Acacia* | E | − | + | + | + | + |
| *Indigofera* | E | + | + | + | + | + |
| *Prosopis* | E | − | + | − | + | − |
| *Tephrosia* | E | − | + | − | − | − |
| Malvaceae | E | + | + | + | + | + |
| *Bacopa* | E | − | − | − | − | − |
| Poaceae | A | + | + | + | + | + |
| *Polygala* | E | − | + | + | − | + |
| *Salvadora* | E,C | + | − | + | − | − |
| *Tamarix* | E | − | − | − | + | − |
| *Typha* | A | − | + | + | + | + |
| *Phyla* | E | − | + | − | − | + |
| *Fagonia* | E | + | − | + | − | + |
| *Tetraena* | E | − | + | − | − | − |
| *Tribulus* | E | + | + | − | + | + |

Gray cells indicate plants recorded in the pollen rain but not in the relevés; other colors indicate their occurrence in their habitats of provenance. Red = Marginal dry wadis; Blue = Wet wadis; Orange = Low plain calcareous shores; Yellow = Coastal sand-sabkhas. The pollination syndromes refer to the flora of Dhofar. A = anemophily; C = chiropterophily; E = entomophily; H = hydrophily; * *Phoenix* pollination is artificial [12,51]; + = presence; − = absence.

## 6. Conclusions

This study has examined five estuaries, four of which—WS, WR, WD, WT—are located in a tract of the Dhofari coast under the influence of the monsoonal circulation. WS and WR host mangrove communities, WD was the site of an ancient settlement and WH is now at the periphery of the sprawling town of Hasik. Human impact at WH is also testified by the presence of a palm grove in a clear state of abandonment. Isolated date palm trees also occur in WD and WT and could be the remnants of the antique introduction of these plants in the area.

By comparing the plant communities growing along the estuaries and the pollen rain from the same sites, it is possible to note that the pollen records offer a broad comprehensive picture of the flora and vegetation of the area, rather than a punctual indication of the plants growing at a short distance. Only a few plants show such low pollen diffusion that their presence in the pollen samples is only recorded if they grow close to the sampling point. Among these is *Avicennia marina*, a plant that plays a key role in the environmental history of the coasts of Dhofar and the Arabian Peninsula, as a whole.

**Supplementary Materials:** The following supporting information can be downloaded at: https://www.mdpi.com/article/10.3390/su141711038/s1, S1: Pollen morphology of plants of the Dhofari coast; S2: Statistical-comparative analysis of the Amaranthaceae pollen grains; S3: Relevés from the estuaries.

**Author Contributions:** C.B., M.M.L. and L.P. are responsible for conceptualization, methodology, analyses, investigation and writing; statistical analyses by F.C.; preliminary floristic survey by R.M.B.; laboratory preparations by T.G. All authors have read and agreed to the published version of the manuscript.

**Funding:** Funding was provided by Centro Studi Erbario Tropicale (CSET - Herbarium FT), Università di Firenze.

**Data Availability Statement:** Voucher specimens of the plants collected for identification purposes are housed at FT herbarium, Centro Studi Erbario Tropicale (https://www.bio.unifi.it/p130.html, accessed on 30 April 2022).

**Acknowledgments:** This research has been carried out within the activities of the Italian Mission to Oman (IMTO) of the University of Pisa (Italy), under the auspices of the Office of the Adviser to His Majesty the Sultan for Cultural Affairs, Sultanate of Oman and logistical assistance from the Office of the Adviser to His Majesty the Sultan for Cultural Affairs, Sultanate of Oman. The authors wish to thank Carlotta Bambi, Asia Bonciani, Laura Tagliapietra and Irene Viviani for their contribution to the study of pollen morphology; Lorella Dell'Olmo of the Università di Firenze (Italy) for drawing the map in Figure 1. A previous version of this work entitled "Palynological Flora Of The Coastal Habitats In Dhofar (Sultanate Of Oman)" was awarded with the best poster prize in pollen biology and structure at the Mediterranean Palynology Societies Symposyum—MedPalynoS-2021, for which we thank Assunta Florenzano and Anna Maria Mercuri. We also thank the reviewers for their comments and improvements to this paper.

**Conflicts of Interest:** The authors declare no conflict of interest.

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
