# Peer review of "Modern Pollen Analysis in the Estuary Habitats along the Coast of Dhofar (Sultanate of Oman)"

_sustainability, doi:10.3390/su141711038_

Round 1

Reviewer 1 Report

This article represents baseline research studying the flora and local pollen rain along the coast of Dhofar. It lets to gain knowledge on regional pollen morphology, pollen diffusion and representation. This kind of studies have proven to be useful and necessary to better interpret paleoenvironmental records from the past, as well as contribute to understand pollen-vegetation relationships in specific environmental and climate conditions. Therefore, this paper constitutes a major contribution to the palynological research in Oman. Despite some minor restructuration is needed, the content of the paper is correct, interesting, and well written. I propose minor reorganization of the text by relocating some paragraphs and creating a new section regarding the geographical setting that should not be in the Introduction. For the Discussion section, I suggest complementary numerical analysis that could reinforce the interpretation of the results, despite the decision is of the authors. Therefore, major strengths of this paper are the novelty and relevance of the research for the study area as well as the important work on pollen morphology, while potential weaknesses are limited integration of author’s interpretation with further modern analogue research. See specific comments in the attached file.

Author Response

Dear Reviewer 1,

Thank you for all your suggestions for improvement. We followed all of your suggestions and integrated them in the paper. The only suggestion we were not able to address is the one regarding complementary numerical analysis in the Discussion. This is because a larger number of sampling sites than ours is necessary for a quantitative assessment of the representation of the pollen taxa through fidelity indices. We added a comment on this in the Discussion.

Kind regards,

The authors.

Reviewer 2 Report

Dear authors, 

This paper is interesting, written well, and organized, however, there are some comments should be considered for improving before publication:

The most important results should be included in the abstract

You should follow the guidelines of the in the formatting and citing the reference in the text

In fig.1, please put an arrow on the top part that shows the areas in the blow part and what is the indication of black arrow in the blow part?

The letters on fig. 2 and 3 should be bold or in a big size

The tables title have to be above the tables        

Author Response

Dear Reviewer 2,

Thank you for your comments. We followed all of your suggestions and integrated them in the paper.

Kind regards,

The authors.

Reviewer 3 Report

This study (Sustainability-1855146, Modern pollen analysis in the estuary habitats along the coast 2 of Dhofar (Sultanate of Oman)) is simply based on investigate the flora and vegetation that develop on the estuaries along the coast of Dhofar and the local pollen rain, with a special focus on the plant distribution and the pollen morphology of the  main components.. Based on the findings, By comparing the plant communities growing along the estuaries and the pollen rain from the same sites, it is possible to note that the pollen records offer a general picture of the flora and vegetation of the area, rather than a punctual indication of the plants growing at a short distance. Only a few plants have such a low diffusion of their pollen that their presence in the pollen samples is only recorded if they grow close to the sampling point. The paper contains new science explaining clearly the new science versus current knowledge? It is written in standard, grammatically correct English including sufficient data, and it has a good structure (introduction, experiment, results and discussion, etc.). 

Author Response

Dear Reviewer 3,

Thank you for your feedback.

Kind regards,

The authors